# PHYSICS-INFORMED NEURAL NETWORKS FOR TRANSFORMED GEOMETRIES AND MANIFOLDS

## ABSTRACT

Physics-informed neural networks (PINNs) effectively embed physical principles into machine learning, but often struggle with complex or alternating geometries. We propose a novel method for integrating geometric transformations within PINNs to robustly accommodate geometric variations. Our method incorporates a diffeomorphism as a mapping of a reference domain and adapts the derivative computation of the physics-informed loss function. This generalizes the applicability of PINNs not only to smoothly deformed domains, but also to lower-dimensional manifolds and allows for direct shape optimization while training the network. We demonstrate the effectivity of our approach on several problems: (i) Eikonal equation on Archimedean spiral, (ii) Poisson problem on surface manifold, (iii) Incompressible Stokes flow in deformed tube, and (iv) Shape optimization with Laplace operator. Through these examples, we demonstrate the enhanced flexibility over traditional PINNs, especially under geometric variations. The proposed framework presents an outlook for training deep neural operators over parametrized geometries, paving the way for advanced modeling with PDEs on complex geometries in science and engineering.

## 1 INTRODUCTION

Physics-informed neural networks (PINNs) (Raissi et al., 2019) are simple yet surprisingly powerful machine learning approaches to incorporate physical knowledge, in particular, formulated as partial differential equations (PDEs), into the training of neural networks. In the burgeoning field of physics-informed machine learning (Karniadakis et al., 2021), they play a significant role due to their versatile applicability in a wide range of problems in science and engineering, for instance, connecting measurement data and known physics in fluid dynamics (Raissi et al., 2020).

Substantial challenges remain, unfortunately, and training PINNs is difficult in practice (Wang et al., 2023). A considerable degree of hyper-parameter tuning, coupled with precise weighting of competing loss terms, is often indispensable to derive satisfactory solutions, irrespective of the challenges associated with determining an effective network architecture or finding a sufficient optimum (Raissi et al., 2019). Training physics-informed neural networks becomes especially challenging in the context of complex problems Krishnapriyan et al. (2021), and accurately adhering boundary conditions is tough (van der Meer et al., 2022).

Distance functions exhibited potential in precisely enforcing boundary conditions (Sukumar & Srivastava, 2022), however, the idea is limited to rather simple geometries where distance functions can be constructed. Addressing the challenges posed by complex geometries, PhyGeoNet (Gao et al., 2021) first proposed the integration of a geometric mapping to accommodate a convolutional neural network architecture to unstructured domains. A geometric mapping has also been used for Fourier neural operators on transformed geometries (Li et al., 2022), but not in a physics-informed manner. Interestingly, hardly any approach of PINNs for manifolds has been proposed, e.g., for problems on surfaces in three-dimensional space, only few works like (Fang et al., 2021), (Costabal et al., 2022) and, to a limited extend, (Bonev et al., 2023) show attempts in this direction.

Drawing upon and meticulously synthesizing prior concepts, we propose a novel yet straightforward approach to facilitate the application of physics-informed neural networks (PINNs) to complex or varying geometries, as well as for solving partial differential equations (PDEs) on manifolds.

Our approach integrates a geometric transformation of a reference domain to represent the computational domain, while concurrently adjusting the derivative computation of the physics-informed loss function. This engenders a latent representation of the solution to the PDE on the reference domain, yielding improved generalization properties across similar geometries and facilitating the implementation of exact boundary conditions. Such a formulation not only extends the applicability of PINNs to smoothly deformed domains, but also lower-dimensional manifolds, and extends its utility to free boundary problems or shape optimization.

The paper is divided into three sections: Initially, we outline the formulation of PDEs on transformed geometries. Subsequently, we situate physics-informed neural networks within the realm of transformations and manifolds. In the final section, through four illustrative examples, we exhibit diverse applications of our novel approach, thereby unveiling exciting new horizons for physics-informed neural networks.

## 2 PROBLEM SETTING

### 2.1 DOMAIN AND TRANSFORMATION

Let us consider a diffeomorphism $\varphi$, a differentiable function with differentiable inverse, with

$$\varphi : \Omega_{\text{ref}} \to \Omega, \tag{1}$$
$$x \mapsto y,$$

that maps an open and bounded $m$-dimensional domain $\Omega_{\text{ref}} \subset \mathbb{R}^m$ to an $m$-dimensional manifold $\Omega \subset \mathbb{R}^n$ embedded in $n$-dimensional Euclidean space. We will refer to $\Omega_{\text{ref}}$ as *reference domain*, for instance, a unit square as illustrated in Figure 1, and to $\Omega$ as *computational domain*, which is the domain where the subsequent problem will be posed.

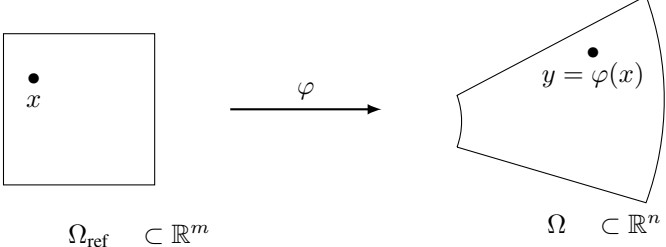

Figure 1: A diffeomorphism $\varphi$ smoothly transforms reference domain $\Omega_{\text{ref}}$ into $\Omega$.

### 2.2 DIFFERENTIAL EQUATION

Consider a generic scalar differential equation in $\Omega$ of the form

$$\mathcal{L}u = f, \qquad \text{in } \Omega, \tag{2}$$
$$u = g, \qquad \text{on } \partial\Omega, \tag{3}$$

where $\mathcal{L}$ is an arbitrary continuous differential operator and $f : \Omega \to \mathbb{R}$ a source term. For simplicity, let us denote Dirichlet boundary conditions $g : \partial\Omega \to \mathbb{R}$ only. We assume that system (2)-(3) is well-posed and has a unique and smooth solution $u$.

Incorporating the transformation $\varphi$ into this system, we distinguish between the following two cases.

### 2.2.1 MANIFOLD: $m < n$

In case of embedded manifolds, $m < n$, the system denoted by equations (2)-(3) needs to be formulated in terms of directional derivatives. This translates to expressing the system in local coordinates

$$\mathcal{L}_x(u \circ \varphi) = f \circ \varphi, \qquad \text{in } \Omega_{\text{ref}}, \qquad (4)$$

$$u \circ \varphi = g \circ \varphi, \qquad \text{on } \partial\Omega_{\text{ref}}, \qquad (5)$$

and provides a straightforward formulation for applying PINNs to problems on manifolds. Notably, without this formulation, solving PDEs on manifolds is not possible — just sampling points along the manifold falls short as this does not compute the directional derivatives along the manifold.

Some examples using this formulation are presented in Section 4.1 and 4.2.

### 2.2.2 TRANSFORMATION: $m = n$

In case of $m = n$, the differential operator can be understood as $\mathcal{L} = \mathcal{L}_y$ with respect to global coordinates, facilitating the interpretation of $\Omega$ as a (parametrized) domain possessing constant measure. In this spirit, system (2)-(3) can be reformulated in terms of $u_{\text{ref}} : \Omega_{\text{ref}} \to \mathbb{R}$ on the reference domain

$$\mathcal{L}_y(u_{\text{ref}} \circ \varphi^{-1}) = f, \qquad \text{in } \Omega = \varphi(\Omega_{\text{ref}}), \qquad (6)$$

$$u_{\text{ref}} \circ \varphi^{-1} = g, \qquad \text{on } \partial\Omega = \varphi(\partial\Omega_{\text{ref}}), \qquad (7)$$

and offers a versatile way of representing solutions on varying geometries.

As $u_{\text{ref}}$ is defined on a latent representation of the geometry, $\Omega_{\text{ref}}$, it changes smoothly with variations in $\varphi$. Moreover, it is beneficial for imposing exact Dirichlet boundary conditions, as they can be stated on a geometrically simple reference domain.

We will demonstrate some applications of formulation (6)-(7) in Section 4.3 and 4.4, after outlining how both formulations can be put into the context of PINNs.

## 3 PHYSICS-INFORMED NEURAL NETWORKS

Physics-informed neural networks are deep neural networks that encode physical effects by training with respect to a loss function incorporating the underlying partial differential equation (Raissi et al., 2019). To approximate the solution of a PDE, let us consider a neural network

$$\hat{u} : \mathbb{R}^n \to \mathbb{R}$$

that maps coordinates $y \in \Omega$ to scalar values $\hat{u}(y) \in \mathbb{R}$. If the network's output depends smoothly on the input coordinates, a differential equation like (2)-(3) can be incorporated into the loss function to guide the update of the weights of the network. Advanced automatic differentiation techniques in modern machine learning frameworks make it feasible to perform the necessary computations, and the universal approximation theorem suggests that we can find a proper solution to our equation.

More technically, sampling a set of $N$ loss points $y_i$ within $\Omega$ and $M$ boundary loss points $z_i$ on $\partial\Omega$, we train the neural network with respect to the loss function

$$MSE(\hat{u}) = \frac{1}{N} \sum_{i=1}^{N} [\mathcal{L}\hat{u}(y_i) - f(y_i)]^2 + \frac{1}{M} \sum_{i=1}^{M} [\hat{u}(z_i) - g(z_i)]^2, \qquad (8)$$

that evaluates the differential operator $\mathcal{L}$ in a point-wise manner and, additionally, penalizes a deviation of our solution from the Dirichlet boundary conditions at the boundary. Provided that the neural network $\hat{u}$ possesses sufficient expressiveness and a sufficient number of loss points is sampled, $\hat{u}$ aspires to approximate the solution $u$ to the PDE.

This approach is simple, yet very powerful, as it not only allows the solution of a wide range of PDEs, but also easily incorporates (noisy) measurement data or can be used to solve inverse problems, bridging the gap between physics-based solvers and data-driven machine learning (Karniadakis et al., 2021).

### 3.1 EXACT BOUNDARY CONDITION WITH OUTPUT TRANSFORM

Dirichlet boundary conditions can be imposed exactly by adding an output transform to the network's outputs (Sukumar & Srivastava, 2022; Lu et al., 2021c). To make the approximate solution satisfy the boundary conditions, we construct the approximation $\hat{u}$ as

$$\hat{u}(y) = \mathcal{N}(y)b(y) + g(y), \qquad y \in \Omega,$$

where $\mathcal{N}$ is the network output, and $b : \Omega \to \mathbb{R}$ is a smooth (distance) function satisfying $b = 0$ at the boundary $\partial\Omega$ and $b > 0$ within $\Omega$. For a unit square, for instance, $b(y) = q(y_1)q(y_2)$ with $q(z) = 4z(1 - z)$ is a viable option. The expressivity of the neural network does not suffer from this output transform, but the approximation $\hat{u}$ satisfies the Dirichlet boundary values $g$ exactly.

However, for complex geometries, formulating a distance function $b$ can pose significant challenges or may even be infeasible. Instead, when writing the problem as a transformation of a simple reference domain, the boundary conditions can be imposed in local coordinates, which makes it relatively easy to impose exact boundary conditions even on complex geometries as we will demonstrate in the following.

### 3.2    PINNs FOR MANIFOLDS

In the manifold case, the PDE has to be rewritten in local coordinates, see formulation (4)-(5), defined over the reference domain. We plug in transformation $\varphi$, mapping from local to global coordinates, as an input feature transform to the network $\hat{u} : \mathbb{R}^n \to \mathbb{R}$, which is defined in global coordinates. Then, we train the transformed network

$$\mathcal{M}(x; \hat{u}) = (\hat{u} \circ \varphi)(x), \qquad x \in \Omega_{\text{ref}},$$

on the reference domain $\Omega_{\text{ref}}$, such that $\hat{u}$ results in an approximation to the solution $u$ of system (4)-(5). Remarkably, Dirichlet boundary conditions can exactly be imposed in this setting by adding an output transform that acts on the local domain. If $b_{\text{ref}} : \Omega_{\text{ref}} \to \mathbb{R}$ is a distance function on the reference domain, we can construct a transformed network as

$$\mathcal{M}(x; \hat{u}) = (\hat{u} \circ \varphi)(x)\, b_{\text{ref}}(x) + (g \circ \varphi)(x), \qquad x \in \Omega_{\text{ref}}, \tag{9}$$

and $\mathcal{M}$ satisfies the Dirichlet conditions $g$ on the boundary of the reference domain exactly.

### 3.3    PINNs FOR TRANSFORMATIONS

In the case of transformations ($n = m$), we solve system (6)-(7) for the differential operator $\mathcal{L}_y$ with respect to global coordinates. For this purpose, we map (loss points from) the reference domain $\Omega_{\text{ref}}$ to global coordinates, and understand $\hat{u} : \mathbb{R}^n \to \mathbb{R}$ as a function in local coordinates. Formulation (6)-(7) motivates to put the transformed network

$$\mathcal{M}(y; \hat{u}) = (\hat{u} \circ \varphi^{-1})(y), \qquad y \in \varphi(\Omega_{\text{ref}}),$$

into the differential operator, which implicitly encodes the derivatives of the transformation. Similar to above, Dirichlet boundary conditions on the reference domain can be imposed by

$$\mathcal{M}(y; \hat{u}) = (\hat{u} \circ \varphi^{-1})(y)\, b_{\text{ref}}(\varphi^{-1}(y)) + g(y), \qquad y \in \varphi(\Omega_{\text{ref}}).$$

In summary, training a neural network $\hat{u} : \mathbb{R}^n \to \mathbb{R}$ with respect to the loss function

$$MSE(\hat{u}) = \frac{1}{N} \sum_{i=1}^{N} \left[ \mathcal{L}\left(\mathcal{M}(y_i; \hat{u})\right) - f(y_i) \right]^2, \quad y_i = \varphi(x_i), \quad x_i \in \Omega_{\text{ref}}, \tag{10}$$

results in an approximation $\mathcal{M}$ to the solution of system (2)-(3) on $\Omega = \varphi(\Omega_{\text{ref}})$, and a latent space solution $\hat{u}$ formulated in terms of local coordinates. This latent $\hat{u}$ depends smoothly on the geometry transformations $\varphi$, and formulation (10) enables the solution of PDEs across complex transformed geometries with exact Dirichlet boundary conditions that are defined by distance functions on a simple reference domain.

In the following section, we will demonstrate the use of both formulations in several examples.

## 4 EXAMPLES

We demonstrate the effectivity of our approach through several representative problems. Through these examples, we highlight the significantly enhanced flexibility offered by our method over traditional PINNs, particularly under geometric variations, unveiling exciting new capabilities for physics-informed neural networks.

The list of examples includes two manifold and two transformation cases:

   (i) Eikonal equation on Archimedean spiral

  (ii) Poisson problem on surface manifold

 (iii) Incompressible Stokes flow in deformed tube

 (iv) Shape optimization with Laplace operator

All examples have been implemented using DeepXDE (Lu et al., 2021b), and the full source code is provided in the supplementary material. Unless stated otherwise, we employ fully connected neural networks with 128 nodes in three hidden layers with $\tanh$ activation function. Optimization is carried out using PyTorch's L-BFGS algorithm over 1000 steps.

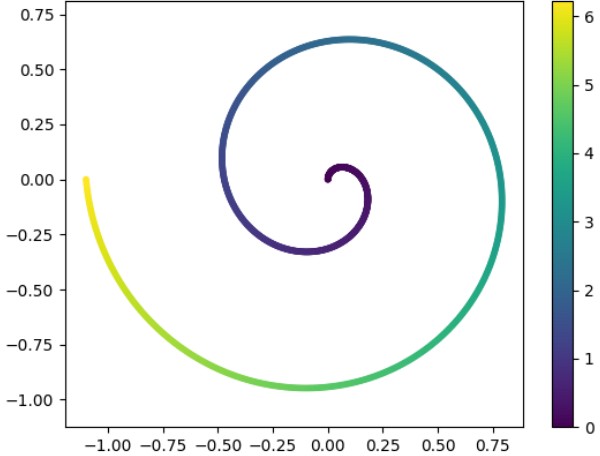

Figure 2: Eikonal equation on Archimedean spiral.

### 4.1 EIKONAL EQUATION ON ARCHIMEDEAN SPIRAL

Our first example serves as a simple benchmark problem to evaluate the accuracy of our method. We consider a one-dimensional manifold $\Omega$ given by a mapping of

$$\Omega_{\text{ref}} = [0, 1] \subset \mathbb{R}$$

to an Archimedean spiral in $\mathbb{R}^2$. Choosing $l = 3.5\pi$, $a = 0.1$ and $r(x) = ax$, it can be written by

$$\varphi(x) = \begin{pmatrix} r(lx) \sin(lx) \\ r(lx) \cos(lx) \end{pmatrix} \in \Omega \subset \mathbb{R}^2.$$

In this domain $\Omega$, we solve the (one-dimensional) Eikonal equation

$$\begin{aligned} \nabla u &= 1, & \text{in } \Omega, \\ u &= 0, & \text{on } (0,0), \end{aligned}$$

and use reformulation (4)-(5) to implement the directional derivatives of the problem, which reads

$$\nabla_x(u \circ \varphi) = 1, \qquad \text{in } \Omega_{\text{ref}},$$
$$u \circ \varphi = 0, \qquad \text{on } \{0\}.$$

This problem can be easily implemented on the basis of (9) and the numerical result is depicted in Figure 2. The maximum value of the solution of this problem corresponds to the length of the spiral, as long as we parameterize the curve by arc length. The length of the spiral is analytically given by

$$\frac{a}{2}(l\sqrt{1+l^2}) + \log(l + \sqrt{1+l^2}) \approx 6.225,$$

and our transformed PINN finds the exact length with an error of $\approx 0.1\%$.

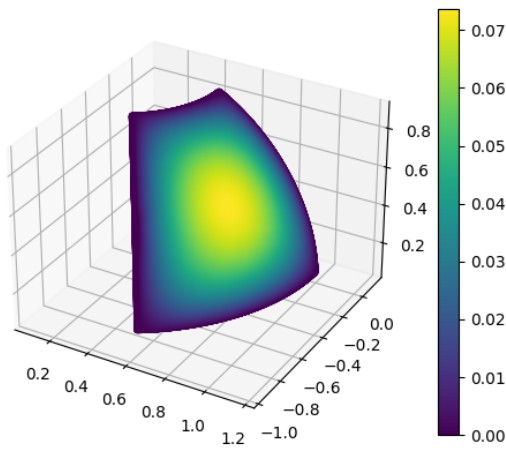

Figure 3: Poisson problem on surface manifold.

## 4.2 POISSON PROBLEM ON SURFACE MANIFOLD

The second example shows how a transformed PINN can solve PDEs on surface manifolds. As a reference PDE, we consider Poisson's equation with uniform Dirichlet boundary conditions

$$-\Delta u = f \qquad \text{in } \Omega, \qquad (11)$$
$$u = 0 \qquad \text{on } \partial\Omega. \qquad (12)$$

The manifold $\Omega$ is chosen as a part of a sphere, written in polar coordinates by

$$\varphi(x_1, x_2) = \begin{pmatrix} \sin(\psi)\cos(\theta) \\ \sin(\psi)\cos(\theta) \\ \cos(\psi) \end{pmatrix}, \quad \begin{array}{l} \psi = x_1 + \psi_0, \\ \theta = x_2 + \theta_0, \end{array}$$

parametrized over the reference domain $\Omega_{\text{ref}} = [0,1]^2$. In our specific example, we choose $f \equiv 1$, $\psi_0 = 0.5$ and $\theta_0 = 1.0$. The resulting manifold and the solution of the problem are depicted in Figure 3. We use reformulation (4)-(5) to express problem (11)-(12) in terms of local coordinates

$$-\Delta_x(u \circ \varphi) = f \circ \varphi, \qquad \text{in } \Omega_{\text{ref}},$$
$$u \circ \varphi = 0, \qquad \text{on } \partial\Omega_{\text{ref}}.$$

and solve this problem as described in Section 3.2 with strong boundary conditions. The function $u : \mathbb{R}^3 \to \mathbb{R}$ is represented by a neural network that maps three-dimensional coordinates, but all derivatives are, due to the formulation in local coordinates, evaluated as directional derivatives along the manifold. To the best of our knowledge, this represents the first instance of a Poisson equation being solved on a manifold utilizing physics-informed neural networks.

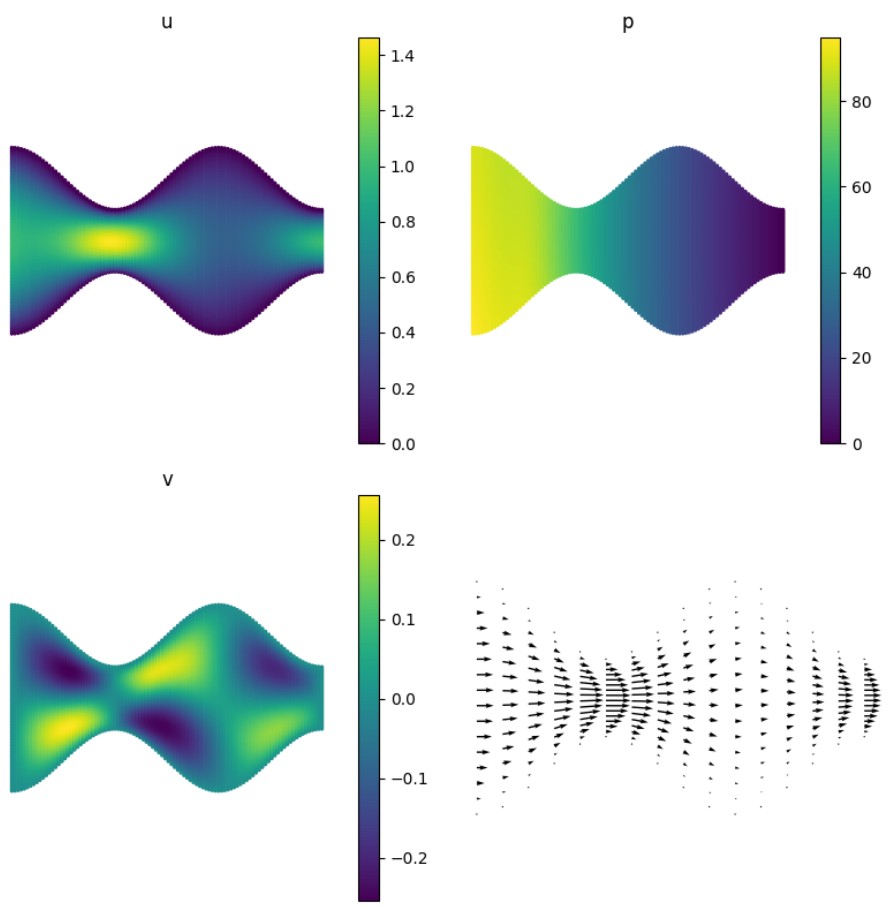

Figure 4: Incompressible Stokes flow in deformed tube.

## 4.3 INCOMPRESSIBLE STOKES FLOW IN DEFORMED TUBE

The third example applies our approach to a physical problem that is ubiquitous in fluid dynamics.

An incompressible, steady-state fluid flow can be described by the Stokes equations

$$-\Delta \mathbf{u} + \nabla p = 0, \qquad \qquad \text{in } \Omega, \qquad \qquad (13)$$
$$\text{div } \mathbf{u} = 0, \qquad \qquad \text{in } \Omega, \qquad \qquad (14)$$

where for $\Omega \subset \mathbb{R}^2$ the unknowns are velocity $\mathbf{u} = (u, v) : \Omega \to \mathbb{R}^2$ and pressure $p : \Omega \to \mathbb{R}$. We examine a flow through a tube from left to right, characterized in terms of (local) Dirichlet boundary conditions on $\Omega_{\text{ref}} = [0, 1]^2$, specifically:

$$u(x_1, x_2) = 4x_2(1 - x_2), \qquad \qquad \text{on } \partial\Omega_{\text{ref}},$$
$$v(x_1, x_2) = 0, \qquad \qquad \text{on } \partial\Omega_{\text{ref}},$$
$$p(x_1, x_2) = 0, \qquad \qquad \text{on } \{1\} \times [0, 1].$$

In our example, we choose a deformed tube $\Omega$ that arises from the transformation

$$\varphi(x_1, x_2) = (x_1, (2x_2 - 1)s(x_1)), \qquad s(x_1) = 0.2 + 0.1\cos(3\pi x_1).$$

The resulting tube geometry and the corresponding flow field are depicted in Figure 4. As we elaborated in Section 3.3, the system defined by (13)-(14) can be solved on $\Omega = \varphi(\Omega_{\text{ref}})$ using exact boundary conditions imposed on the reference domain, and we train the network for 5000 steps.

The numerical solution exhibits the anticipated pattern, wherein the fluid velocity significantly increases at the narrow segment of the tube, accompanied by a more rapid pressure decrement. It's noteworthy that all boundary conditions are met exactly, obviating the need for hyper-parameters to strike a balance between inner and boundary loss.

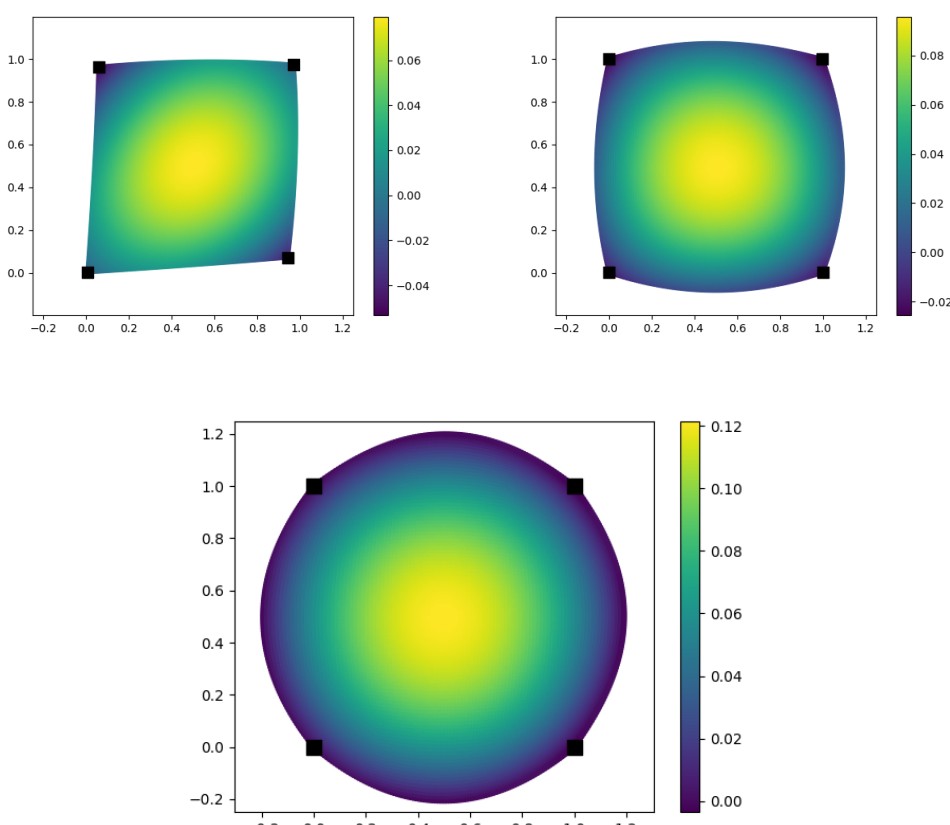

Figure 5: Shape optimization with Laplace operator. Initial configuration, one intermediate step, and final solution. The four black squares indicate the constrained points of the transformation.

## 4.4 SHAPE OPTIMIZATION WITH LAPLACE OPERATOR

Our final example reveals novel capabilities engendered by our approach of representing computational domains via transformations of a reference domain. Indeed, this methodology facilitates free boundary problems and shape optimization in a straightforward manner. Topology optimization has previously been executed using PINNs (Lu et al., 2021c), but with a quantity of interest for the inverse design problem embedded within the PDE problem.

Let us represent the transformation $\varphi$ itself by a neural network

$$\hat{\varphi} : \mathbb{R}^m \to \mathbb{R}^n, \quad x \mapsto y,$$

which can be learned independently (or concurrently) during the training of a physics-informed neural network. Note that, in general, $\hat{\varphi}$ will not be a diffeomorphism, but at least we can assure that it is smooth and differentiable.

The newfound flexibility can be demonstrated in the following example, employing the Laplace operator. Having two neural networks $\hat{u} : \mathbb{R}^2 \to \mathbb{R}$ and $\hat{\varphi} : \mathbb{R} \to \mathbb{R}$, we solve problem

$$-\Delta \hat{u} = 1, \qquad \qquad \text{in } \Omega = \hat{\varphi}(\Omega_{\text{ref}}), \qquad (15)$$

$$\hat{u} = 0, \qquad \qquad \text{on } \partial\Omega = \hat{\varphi}(\partial\Omega_{\text{ref}}), \qquad (16)$$

where the computational domain $\Omega$ is a mapping of $\Omega_{\text{ref}} = [0,1]^2$ via $\hat{\varphi}$. We impose Dirichlet conditions weakly by adding a penalization term to the loss, as in (8), and train both neural networks simultaneously to minimize this PINN loss. Additionally, we fix four points of domain $\Omega$ which imposes constraints to the transformation $\hat{\varphi}$, namely $\hat{\varphi}(x) = x$ for $x \in \{(0,0), (1,0), (0,1), (1,1)\}$. In our implementation, both neural networks have a single hidden layer with 1024 nodes, and we execute training simultaneously until convergence which occurs after 18 steps of optimization.

We do not introduce any explicit objective to the optimization, yet observe that the loss attains minimization when the domain manifests as a circle, as illustrated Figure 5. This outcome suggests that weakly imposed boundary conditions pose a distinctive challenge for PINNs in non-convex geometries. It shows how the parametrization of geometries via neural network can address an entirely new class problems, in particular, free boundary problems or shape optimization.

## 5 CONCLUSION AND OUTLOOK

In this paper, we introduced a novel formulation of physics-informed neural networks for transformed geometries, thereby expanding their applicability to domain transformations, manifolds, and free boundary problems. Our methodology opens new pathways for employing PINNs on complex geometries and enables the precise enforcement of Dirichlet boundary conditions via distance functions on intricate geometries. The versatility of our approach was demonstrated through four illustrative examples, encompassing applications ranging from solving the Eikonal to the Poisson equation on manifolds, analyzing incompressible fluid flow within a parametrized tube geometry, to applying our framework to free boundary problems where the optimal domain geometry itself is learned by a neural network.

The proposed framework presents an outlook for training physics-informed neural operators, like DeepONets (Lu et al., 2021a) or Fourier Neural Operators (Li et al., 2022), on parametrized geometries as the latent representation of the solution on the reference domain can generalize well between similar transformations. It paves the way for advanced modeling with PDEs on complex geometries in science and engineering.

## REPRODUCIBILITY STATEMENT

All numerical examples presented in this paper have been kept straightforward for ease of understanding and have been thoroughly implemented to ensure correctness. The full source code of the examples is self-contained and it is provided in the supplementary material for independent reproduction and analysis.

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
