# OpenReview forum: "Physics-informed neural networks for transformed geometries and manifolds"
_ICLR.cc/2024/Conference — ICLR 2024 Conference Withdrawn Submission_

### Official Review · Reviewer_2tfv · 2023-10-28

**Soundness:** 2 fair
**Presentation:** 1 poor
**Contribution:** 2 fair
**Rating:** 3
**Confidence:** 4

**Summary:**

The paper intends to improve the performance of PINN on domains of complex geometries. The method is to  use smooth transformations to transform a complex geometry to less complex one which is a called reference domain. If the transformations are differentiable, the training of modified PINNs is the same as training vanilla PINNs.

**Strengths:**

**Originality:** The paper implements diffeomorphisms to the problem of PINN on complex geometries.

**Quality:** The paper explores the proposed methods on some typical examples to demonstrate the effectiveness of the method.

**Clarity:** The idea is conveyed directly and straightforward.

**Significance:** Combining diffeomorphism with training of neural network is somewhat interesting and natural, due to the differentiability of transformations.

**Weaknesses:**

One of the major weakness is that the paper does not include experiments of comparison between modified PINN and vanilla PINN. In order to show the effectiveness of the proposed method, the author should also test the performance of PINN on all the problems in section 4.

**Questions:**

If the original problems $L(u)=f$ in $\Omega$ is transformed to $L_x(u \circ \phi) = f$ on reference domain $\Omega_{ref}$, then $L_x$ should not equal $L$. The calculation of $L_x$ should use chain rule. In your paper, this part is hardly touched. How did you actually implement your method in experiments?

---

> ### Author Response · Authors · 2023-11-23
>
> Thank you for your review. We would like to address the following comments.
>
> _One of the major weakness is that the paper does not include experiments of comparison between modified PINN and vanilla PINN. In order to show the effectiveness of the proposed method, the author should also test the performance of PINN on all the problems in section 4._
>
> Unfortunately, we do not understand how vanilla PINNs should be applied to the case of manifolds, as directional derivatives have to be computed.  It might be possible to elaborate the differential operators (e.g., Laplace-Beltrami) explicitly and compute those in a “vanilla” way, but this is what our approach implicitly does, formulated in local coordinates, putting all the work into the automatic differentiation.
> In the case of equi-dimensional transformations, vanilla PINNs could actually be applied and would lead to a well-studied reference solution. Thank you for this suggestion.
>
>
> _If the original problems $L(u)=f$ in $\Omega$ is transformed to $L_x(u \circ \phi) = f$ on reference domain $\Omega$ ref, then $L_x$ should not equal $L$. The calculation of $L_x$ should use chain rule. In your paper, this part is hardly touched. How did you actually implement your method in experiments?_
>
> In the manifold case, chain rule applies and is carried out by the automatic differentiation framework.
> In the transformation case, we explicitly neglect the chain rule. It’s somewhat arbitrary (and we are sorry if we didn’t make this sufficiently clear), but this idea leads to general transformed domains.
> Regarding the actual implementation: We provide our full source code as supplementary material and refer to it for all implementation details.

---

> > ### Comment · Reviewer_2tfv · 2023-11-23
> >
> > Thanks for clarifying. I agree that this paper has novelty in solving PDE problems on manifold. However, regarding geometry transformation, here's my follow-up question:
> >
> > I'm still confused with eq. (6)-(7). Given $\mathcal{L}(u_{ref})=f$, how can $u=u_{ref}\circ \phi^{-1}$ satisfy $\mathcal{L}(u)=f$, as you mentioned above eq. (6)-(7) that $\mathcal{L}=\mathcal{L_y}$? In my understanding, generally $u$ satisfies $\mathcal{L_y}(u)=f$ and $\mathcal{L}\neq \mathcal{L_y}$.
> >
> > I did look at the code, but due to this fundamental question I didn't follow.

---

> > > ### Author Response · Authors · 2023-11-23
> > >
> > > In (6)-(7) we only substitute $u = u_{ref} \circ \phi^{-1}$ into (2)-(3), and take $\mathcal{L} = \mathcal{L}_y$ as it is in (2)-(3) (global coordinate $y \in \Omega$). The reformulation is essentially trivial, but it is quite interesting because $\phi$ now governs the shape of domain $\Omega$.

---

### Official Review · Reviewer_UpPb · 2023-10-29

**Soundness:** 3 good
**Presentation:** 3 good
**Contribution:** 3 good
**Rating:** 6
**Confidence:** 4

**Summary:**

In this paper, it is argued that the existing approaches to physics-informed neural networks are not apt for complex and transforming geometries. To this end, the paper presents an approach to introduce geometric transformation within the physics-informed neural network design. Concretely, it enforces the Dirichlet boundary condition using distance function to account for complex geometries. Experimental results on four different examples are shown to demonstrate the suitability of the method.

**Strengths:**

* It is a well-written paper.
* The use of Dirichlet boundary conditions is promising.
* An initial approach to explore a new direction for more promising neural network design.

**Weaknesses:**

* Some of the technical notations are not fully exposed and detailed.
* Experiments are limited on toy-example and missing on the manifolds which are widely used in science and engineering application.
* The paper misses to highlight the limitations of the proposed approach.


Kindly refer to the Questions section for more comments.

**Questions:**

## Domain and Transformation

It’s better to include the dimension of the variables on the side of the Eq(1).

## 2.2.1 Manifold: $m < n$

$\mathcal{L}_x$ and $\mathcal{L}_y$ need more explanation. The subscripts have not been explained. Diagram conveys that one is in the reference domain and other is in the computational domain yet it's better to write near the equation (4)-(5) and following equation.

## 3.1 Exact boundary condition with output transform

Kindly help me understand the approximation of $\hat{u}$, given that the inverse must hold and the proposed approximation is not linear.

## 4.4 Shape Optimization with Laplace Operator
I am not entirely convinced with the imposed boundary condition. What could be considered a weak boundary condition is not fully exposed in the paper. Furthermore, I request the authors to perform some experiments and analysis of the proposed theory on negative curvature surfaces with the introduced local approach. Also, the use of Laplace-Beltrami operator for shapes.

In addition to the above, experiment on Low-Dimensional manifolds is simple and not convincing to me for real application. I request the authors to provide some analysis and results on popular manifolds such as low-dimensional SPD, Grassmannian manifolds, etc.

---

> ### Author Response · Authors · 2023-11-23
>
> Thank you for your review. We would like to answer your comments and questions as follows.
>
> $\mathcal{L}_x$ _and_ $\mathcal{L}_y$ _need more explanation. The subscripts have not been explained. Diagram conveys that one is in the reference domain and other is in the computational domain yet it's better to write near the equation (4)-(5) and following equation._
>
> Thanks for your feedback, other reviewers addressed the same issue. We only stated “$\mathcal{L} = \mathcal{L}_y$ with respect to global coordinates” and assumed that it’s clear that the subscript indicates the derivation variable. Obviously, this it is not the case and the subscript needs more explanation.
>
> _Exact boundary condition with output transform: Kindly help me understand the approximation of $\hat u$, given that the inverse must hold and the proposed approximation is not linear._
>
> As the smooth distance function $b$ is zero at the boundary, $N(y) u(y)$ is zero on the boundary, and, therefore, $\hat u(y) = g(y)$ satisfies the Dirichlet boundary values $g(y)$ at the boundary.   Unfortunately, we do not understand what you are referring to with ‘inverse must hold’, and why the approximation should to be ‘linear’.
>
> _I am not entirely convinced with the imposed boundary condition. What could be considered a weak boundary condition is not fully exposed in the paper._
>
> A weak boundary condition (in the context of PINNs) means imposing the boundary condition by adding a penalizing loss term, as outlined in (8). You noted correctly that we missed introducing this wording in the context of (8) and it should be added.
>
> _Experiments are limited on toy-example and missing on the manifolds which are widely used in science and engineering application._
>
>   We tried to demonstrate our method on minimal working examples to make the setup and implementation clearly understandable. It’s unfortunate if you consider them as too limited, and we will take this feedback into account.
>
> _The paper misses to highlight the limitations of the proposed approach._
>
> Unfortunately, that is true, we will add a paragraph on this.
>
> _Furthermore, I request the authors to perform some experiments and analysis of the proposed theory on negative curvature surfaces with the introduced local approach. Also, the use of Laplace-Beltrami operator for shapes._
>
> We do not see why negative curvature should have any relevant impact on our method, the proposed method also works with negative curvature surfaces.  Also, our second example demonstrates a Poisson problem on a part of a sphere, which - as we formulate the derivatives in local coordinates - corresponds to a Laplace-Beltrami on the manifold. Our apologies that we didn’t point this out explicitly.
>
> _In addition to the above, experiment on Low-Dimensional manifolds is simple and not convincing to me for real application. I request the authors to provide some analysis and results on popular manifolds such as low-dimensional SPD, Grassmannian manifolds, etc._
>
> Which PDEs are commonly formulated on low-dimensional SPD or Grassmannian manifolds?

---

### Official Review · Reviewer_VH6u · 2023-10-31

**Soundness:** 2 fair
**Presentation:** 2 fair
**Contribution:** 2 fair
**Rating:** 3
**Confidence:** 3

**Summary:**

This paper employs physics-informed neural networks (PINNs) for addressing intricate or changing geometrical configurations. The primary technical innovation lies in the incorporation of a geometric transformation (diffeomorphism) of a reference domain to describe the computational domain.

**Strengths:**

The problem is well defined and the author proposes a clear formulation in solving the problem.

**Weaknesses:**

Unfortunately,  it appears that the problem tackled in the paper is somewhat incremental, and the proposed solution lacks a surprising or profound aspect. In the context of an ICLR paper, I'm seeking a novel problem that has not previously been successfully addressed, made attainable through this approach, or a novel method to solve a well-established problem that has been extensively explored. Unfortunately, neither of these elements seems to be present in the paper.

Furthermore, the examples provided mainly consist of small-scale 2D toy examples. To comprehensively assess the efficacy of this approach, it would be necessary for the authors to set up larger-scale problems that are well-documented in CFD/JCP/CMAME papers.

**Questions:**

How does this work compare with Bonev+ ICML 2023? These authors propose a neural PDE approach using spherical coordinate. Your paper seems to be more general. Can you reproduce some of the examples in their paper so we can have an apple to apple comparison?

---

> ### Author Response · Authors · 2023-11-23
>
> Thank you for your review! Let us address your comments as follows.
>
> _Unfortunately, it appears that the problem tackled in the paper is somewhat incremental, and the proposed solution lacks a surprising or profound aspect. In the context of an ICLR paper, I'm seeking a novel problem that has not previously been successfully addressed yet, made attainable through this approach, or a novel method to solve a well-established problem that has been extensively explored. Unfortunately, neither of these elements seems to be present in the paper._
>
> Thank you for your valuable feedback. We agree that one can consider the problem tackled somewhat incremental, and that our work does not propose a novel method to solve a well-studied problem.
>
> We tried to introduce the approach of including a diffeomorphism within PINNs as a (novel) general concept, and we believe it could benefit from a common introduction. Furthermore, we consider the application of PINNs to manifolds as a problem that has not been successfully addressed, which is made attainable through this approach.
>
> A surprising aspect of our approach is that - in the transformation case - a latent representation of the PDE solution on the reference domain arises. This is likely to improve generalization capabilities for parametrized geometries, but we recognize that we were not able to demonstrate this effectively on a well-studied problem.
>
>
> _Furthermore, the examples provided mainly consist of small-scale 2D toy examples. To comprehensively assess the efficacy of this approach, it would be necessary for the authors to set up larger-scale problems that are well-documented in CFD/JCP/CMAME papers._
>
> As mentioned above, we tried to introduce the approach as a general concept and, therefore, we set up easily understandable examples that demonstrate the efficacy of the approach. We tried to choose examples with a clear setup and analytical solutions, accompanied by a transparent and manageable implementation.
>
> However, we accept that there is a request for applying our method to larger-scale, well-documented problems that would comprehensively assess the efficacy for non-toy examples.
>
>
> _How does this work compare with Bonev+ ICML 2023? These authors propose a neural PDE approach using spherical coordinate. Your paper seems to be more general. Can you reproduce some of the examples in their paper so we can have an apple to apple comparison?_
>
> Bonev+2023 propose Spherical Fourier Neural Operators (Spherical FNOs) that put the concept of FNOs efficiently onto spheres. As they use spherical harmonics for the Fourier transform, their approach is limited to spheres by definition.  Our approach is more general in the sense that we can model arbitrary manifolds. We think an apple-to-apple comparison doesn’t make sense, because it would be restricted to the case of spheres where the SFNOs will clearly outperform all aspects of our method (accuracy and speed) as it is explicitly tailored to this case.

---

### Official Review · Reviewer_2kyY · 2023-10-31

**Soundness:** 2 fair
**Presentation:** 2 fair
**Contribution:** 1 poor
**Rating:** 1
**Confidence:** 4

**Summary:**

The work proposes a method to enhance Physics-informed Neural Networks (PINNs) by integrating geometric transformations, to address challenges posed by complex or non-euclidean geometries.
The method utilizes a diffeomorphism $\phi$ that maps a reference domain $\Omega_{ref}$ to the observation domain $\Omega$, adapting the derivative computation in the physics-informed loss function. The approach was demonstrated through various problems: Eikonal equation on Archimedean spiral, Poisson problem on surface manifold, Incompressible Stokes flow in deformed tube. Finally, they show that their method can be applied to perform shape optimization according to a Laplace PDE loss.

**Strengths:**

The paper is easy to read and the geometric transformation seems reasonable to solve this kind of problem. The first three different examples each test a different geometric setting. The figures are pretty.

**Weaknesses:**

The method relies on the output transformation trick to enforce boundary conditions (BC), which is well suited for Dirichlet BC only. It would not be applicable as is for different kinds of BC, but the authors have a much more general claim.

Except for the last example, which we will discuss next, the diffeomorphism $\phi$ is known a priori. Therefore the method in such case simply looks like a change in variable with a known function. How can you apply this method on a domain which is not equipped with such a transformation ?

The last example is very mysterious to me. I actually do not understand what the method is supposed to achieve by learning simultaneously to impose the PDE constraint and the geometric transformation. Do we know what target geometry the network should converge to ? Besides, the network that learns the transformation is not a diffeomorphism, so there is no guarantee that the optimization problem finds a correct solution.

The authors do not compare their method with any existing work. There is no literature review. As a result, we do not really understand why these problems cannot be tackled with existing methods. Why do they fail ?

The authors do not provide any numerical results for their methods, and even the qualitative results do not include the ground truth solutions. It is therefore impossible to judge the effectiveness of the method.

**Questions:**

What is the difference between $\mathcal{L}$, $\mathcal{L}_x$ and $\mathcal{L}_y$ concretely for each example ?

 What does the following sentence mean ? "transformed PINN finds the exact length with an error of = 0.1 \%" .

---

> ### Author Response · Authors · 2023-11-23
>
> Thank you for your review! Allow us to respond to your comments as follows.
>
> _The method relies on the output transformation trick to enforce boundary conditions (BC), which is well suited for Dirichlet BC only. It would not be applicable as is for different kinds of BC, but the authors have a much more general claim._
>
> We are not sure which claim you are referring to. Does this refer to “For simplicity, let us denote Dirichlet boundary conditions only”?   We are not claiming that our approach is applicable to different kinds of BC than Dirichlet ones, and we definitely didn’t intend to let it appear more general. Besides, regarding Neumann boundary conditions, one can use the domain transformation to compute the outer normal of the transformed geometry, but it’s unclear how to strongly impose Neumann boundary conditions, as it’s not clear for PINNs in general.
>
> _Except for the last example, which we will discuss next, the diffeomorphism is known a priori. Therefore the method in such case simply looks like a change in variable with a known function. How can you apply this method on a domain which is not equipped with such a transformation?_
>
> In our work, we assume that the diffeomorphism is given, and we assume that the domain geometry is defined by the mapping of the reference domain via the diffeomorphism. If, vice versa, a geometry is given, e.g., by a mesh, the diffeomorphism might be learned, but it’s definitely an open question how this would be achieved effectively.
>
> _The last example is very mysterious to me. I actually do not understand what the method is supposed to achieve by learning simultaneously to impose the PDE constraint and the geometric transformation. Do we know what target geometry the network should converge to ? Besides, the network that learns the transformation is not a diffeomorphism, so there is no guarantee that the optimization problem finds a correct solution._
>
> In the last example, we tried to make the mapping more general by using an NN as transformation (targeting the concern of the limitation that the diffeomorphism has to be known a priori, as in your previous comment).
> It was intended as an explorative example to show what could be done incorporating a parametrized transformation.
> We believe that we were able to demonstrate that this idea results in a geometrically very flexible PDE solver, e.g., in contrast to classical FEM solvers that would require re-meshing.
>
> However, as you mentioned (and we stated clearly), the network has no guarantee to be diffeomorph and this stretches the framework we outlined beforehand beyond its assumptions.
>
> We agree that our objective function - not including an explicit target - is somewhat arbitrary and that we are not able to state analytically what the network should converge to. Experience with PINNs suggests that a convex shape is easiest to optimize for w.r.t. weak boundary conditions, and this is what our example actually results in.
>
> As this example opens too many concerns, we conclude in removing it completely upon further investigation.
>
> _The authors do not compare their method with any existing work. There is no literature review. As a result, we do not really understand why these problems cannot be tackled with existing methods. Why do they fail ?_
>
> We recognize that our examples need more comparisons with similar problems from literature.   Unfortunately, to the best of our knowledge, we are not aware of previous works that applied PINNs to manifolds, which severely limits the possibilities of a comparison as well as a literature review.
>
> _The authors do not provide any numerical results for their methods, and even the qualitative results do not include the ground truth solutions. It is therefore impossible to judge the effectiveness of the method._ /
> _What does the following sentence mean? "transformed PINN finds the exact length with an error of = 0.1 %”_
>
> The first example is constructed in a way that the numerical solution can be compared to an analytical solution. We were not providing a plot for both solutions, because they are trivial and closely aligned s.t. a plot didn’t make sense. Instead, we stated that the numerical solution fits the analytical solution “with an error of 0.1%”. This is what the sentence meant, sorry for the incomprehensible expression.
>
> We acknowledge that our work is lacking analytical results for example 2 and a comparative study for example 3.
>
> _What is the difference between L, L_x and L_y, and concretely for each example ?_
>
> The subscript indicates the derivation variable of the differential operator. As other reviewers addressed the same issue, we have to make this more explicit along with equations (4)-(5).   The examples are split into manifold and transformation cases, where manifold corresponds to L_y and transformation to L_x. We will provide more clarity here.

---

> > ### Comment · Reviewer_2kyY · 2023-11-23
> > **Acknowledge**
> >
> > Thank you for your response, I will keep my score.